# An Effective Hybrid Fungicide Containing Tea Tree Oil and Difenoconazole for Grape Powdery Mildew Management

**Moshe Reuveni [1,2,\*], Cristobal J. Arroyo [1] and Shmuel Ovadia [3]**

1   STK, Bio-Ag Technologies, Ltd., Petah Tikva 4951447, Israel; cristobal@stk-ag.com
2   Shamir Research Institute, University of Haifa, Katzrin 1290000, Israel
3   Shahaf, Karmey Yosef 9979700, Israel; shmovadia@gmail.com
\*   Correspondence: moshe@stk-ag.com

**Abstract:** Grape powdery mildew caused by *Erysiphe necator* (Schw.) Burr. is a destructive disease in vineyards. Synthetic fungicides are the main tool to combat this disease. The search for new alternatives to reduce pesticide usage and tactical approaches for resistance management encouraged us to develop the novel strategy that we report here. We evaluated the efficacy of a new premixed hybrid fungicide containing the demethylation inhibitor (DMI) difenoconazole and essential tea tree oil (TTO), derived from the *Melaleuca alternifolia* plant, against grape powdery mildew in seven field trials and two large-scale demonstration trials conducted in two different regions in the world, including Chile and Israel. Foliar sprays of difenoconazole-TTO were applied as a preventive treatment in field trials at 40–80 up to 80–160 gr/ha active ingredient, and they were highly effective in controlling powdery mildew on the fruit clusters of both wine and table grapes in experimental and large-scale demonstration trials and provided up to 99% efficacy in disease incidence and severity compared with the untreated control. Difenoconazole-TTO was as or more effective than other DMI fungicides, including difenoconazole, a pre-mixed fungicide boscalid-pyraclostrobin, or treatments that included various fungicides applied in rotation or mixtures of fungicides. The results suggest that a combination of difenoconazole-TTO with a reduced synthetic chemical load can be included in powdery mildew control programs for grapevine as a strategic approach in fungicide resistance management in vineyards.

**Keywords:** powdery mildew control; *Erysiphe necator*; *Melaleuca alternifolia*; resistance management; *Vitis vinifera*

## 1. Introduction

Grapevine is one of the most important crops worldwide and grape powdery mildew disease caused by the fungal pathogen *Erysiphe necator* (syn. *Uncinula necator*) (Schw.) Burr. is a destructive disease in vineyards worldwide. It is a significant disease in terms of expenses for control and losses in the quality and yield of wine, juice, raisins, and table grapes worldwide [1,2]. Disease management relies mainly on using multiple sprays per season of protectant and systemic fungicides such as strobilurins, quinone outside inhibitors (QoIs), demethylation inhibitors (DMIs), succinate dehydrogenase inhibitor (SDHI), and sulfur [3–7]. However, strains of *E. necator* with reduced sensitivity to QoIs and DMIs, and to metrafenone, have been reported worldwide in vineyards [3,6,8–11]. A cross-resistance among the DMI fungicides has been shown [12]. Re-application of ineffective fungicides enhances the risk of reinforcing a resistant population [13]. Therefore, new, and effective approaches to existing products that provide different and multiple modes of action with a lower risk for fungicide resistance are required to increase options for the control of plant diseases [4].

The essential Australian tea tree oil (TTO) derived from the *Melaleuca alternifolia* plant contains many components, primarily terpenes, sesquiterpenes, and their respective

alcohols [14]. It contains up to 15% of 1,8-cineole and at least 30% of terpinen-4-ol, which are the main active constituents of TTO [14]. TTO has been shown to be an effective antiseptic bactericide [14] and fungicide [15–18] due to its ability to inhibit respiration and disrupt membrane permeability [19,20].

A new concept of a hybrid fungicide has been introduced to reduce the fungicide resistance risk and chemical load on the environment. A pre-mixture containing the DMI difenoconazole (200 g/L) and tea tree oil (400 g/L) was developed. This study was undertaken to evaluate the efficacy of this new hybrid fungicide compared with other registered synthetic fungicides or fungicide mixtures in controlling grape powdery mildew in vineyards as a tool in resistance management. Preliminary results have been published [21].

## 2. Materials and Methods

### 2.1. Fungicides

The hybrid fungicide difenoconazole-TTO, containing 400 g/L of essential tea tree oil derived from the *Melaleuca alternifolia* plant (FRAC code BM01) and 200 g/L of difenoconazole (FRAC code 3) (Regev 600 EC, STK Bio-Ag Technologies, Petach Tikva, 4951447 Israel), was examined in all trials. The fungicides and fungicide mixtures used in the field studies are provided in Table 1. The rates used in the field studies were based on the fungicide labels or were otherwise recommended by the registrants. The rates of fungicides applied in the field trials are given in active ingredients/ha.

**Table 1.** Fungicides used in this study.

| Active Ingredient | Product Name | G Active Ingredient per Liter | Formulation [1] | Producer | Fungicides [2] Group | FRAC [3] Code |
|---|---|---|---|---|---|---|
| Boscalid plus pyraclostrobin | Bellis | 25.8 and 12.8 | WD | BASF | SDHI and QoI | 11 and 7 |
| Cyflufenamid | Netz | 50 | EW | Nippon Soda | Phenyl-acetamide | U6 |
| Difenoconazole | Score | 250 | SC | Syngenta Crop Protection | DMI | 3 |
| Difenoconazole plus tea tree oil | Regev | 200 and 400 | EC | STK Bio-Ag Technologies | DMI and multi-site | 3 and BM 01 |
| Meptyldinocarp plus Myclobutanil | | 113 and 48 | SC | Dow AgroSciences | Dinitrophenylcrotonates and DMI | 29 and 3 |
| Proquinazid | Talius | 200 | EC | DuPont | Azanaphthalenes | 13 |
| Pyriofenone | Cassuri | 300 | SC | ISK | Aryl phenyl ketone | 50 |
| Quinoxyfen | Abir | 250 | SC | Dow AgroSciences | Azanaphthalenes | 13 |
| Tebuconazole | Folicur | 250 | EC | Bayer CropScience | DMI | 3 |
| Tetraconazole | Domark | 100 | ME | Isagro S.p.A | DMI | 3 |

[1] EC = emulsifiable concentrate, EW = oil-in-water emulsions, ME = microemulsion, SC = suspension concentrates, and WD/WDG = water-dispersible granules. [2] DMI, demethylation inhibitor; QoI, quinone outside inhibitor; SDHI, succinate dehydrogenase inhibitor; and [3] Fungicide Resistance Action Committee.

### 2.2. Field Studies on Grape Powdery Mildew

A total of 2 trials, which included wine and table grape cultivars susceptible to grape powdery mildew, were conducted in Chile in 2017–2018, and 5 field trials and two large-scale demonstration trials were conducted in commercial vineyards of wine and table grapes in the Judean foothills region of Israel during 2020 and 2021. The details of each trial are summarized in Table 2.

**Table 2.** Field trials conducted to assess the efficacy of fungicides and difenoconazole-TTO [a] against grape powdery mildew caused by *E. necator*.

| | | | | | | |
|---|---|---|---|---|---|---|
| **Trials in Chile** | | | | | | |
| **Year** | **Location** | **Cultivar** | **Vineyard Age** | **Number of Sprays** | **Beginning of Sprays** [b] | **End of Sprays** [b] |
| 2017–2018 | Requinoa | "Crimson Seedless"© | 10 | 6 | 26 November 2017 | 8 January 2018 |
| 2017–2018 | Casablanca | "Chardonnay"© | 7 | 5 | 30 November 2017 | 12 January 2018 |
| **Trials in Israel** | | | | | | |
| **Year** | **Location** | **Cultivar** | **Vineyard Age** | **Number of Sprays** | **Beginning of Sprays** | **End of Sprays** |
| 2020 | Gimzo, Judean Foothills, Israel | "Carignan"© | 14 | 4 | April 6 | June 6 |
| 2020 | Eshtaol, Judean Foothills, Israel | "Carignan"© | 9 | 5 | April 6 | June 8 |
| 2020 | Neta Farm, Judean Foothills, Israel | "Carignan"© | 20 | 2 | April 4 | June 3 |
| 2021 | Noham, Judean Foothills, Israel | "Carignan"© | 9 | 3 | April 13 | June 8 |
| 2021 | Gimzo, Judean Foothills, Israel | "Red Loosh"© | 3 | 4 | May 7 | June 17 |
| 2021 [c] | Anava, Judean Foothills, Israel | "Carignan"© | 20 | 2 | May 5 | May 19 |
| 2021 [c] | Latroon, Judean Foothills, Israel | "Muscat of Alexandria"© | 11 | 3 | May 12 | June 17 |

[a] TTO, tea tree oil; [b] details on growth stages (BBCH) for the beginnings and ends of the applications of fungicides in each trial are given in each appropriate table; and [c] large-scale demonstration trials.

### 2.2.1. Field Trials in Chile

The first trial was carried out in the Cacachopal Valley region of Requinoa using the cultivar "Crimson Seedless"© table grapes grown using the Spanish trellis system, with spacing between the rows and vines of 3.5 × 2 m. The climate is warm temperate with winter rains and a prolonged dry season (7 to 8 months), with Mediterranean characteristics (hot and dry summers and rainy, cool, and humid winters) and an annual rainfall of 350–400 mm. A total of 6 foliar sprays were applied, with an average spray volume of 1200 L/ha, at 10-day intervals from 27 November 2017 (BBCH 69) until 16 January 2018 (BBCH 80). The second trial was carried out in the Casablanca Valley using the cultivar "Chardonnay"© wine grapes grown using the Double Guyot system, with spacing between the rows and vines of 2.5 × 1.5 m. The climate conditions are characterized as a cool Mediterranean climate, with pronounced maritime influence and an annual rainfall of 300 mm. A total of 5 foliar sprays were applied at average spray volumes of 600 L/ha at 14-day intervals from 30 November 2017 (BBCH 69) until 25 January 2018 (BBCH 80). Unsprayed vines served as the control. The foliar sprays for both trials were performed using a Maruyama MS073D gun sprayer (pressure 20 bar) (Chiyoda-ku, Tokyo, Japan).

### 2.2.2. Field Trials and Climate Description in Israel

A total of 5 field trials and two large-scale demonstration trials were conducted on susceptible grape cultivars: "Carignan"©, "Red Loosh"©, and "Muscat of Alexandria"©. The vineyards were in the Judean Foothills regions of Israel (Table 2). The wine grapes were spur-pruned with vertical shoot positioning (VSP) and the table grapes were grown as a double veranda (DV). The spacing between the rows and vines was either 4 × 2 m or 3 × 1.5 m, respectively. The fungicides were applied to the vineyards at recommended

rates according to phenological stages and the decision support system (DSS) "Eshkol", which led to a protocol for disease control in Israel [22]. This included applications based on three phenological periods. The first application period (including one or two prophylactic sprays) was in the early season from the first leaf unfolded stage (BBCH-11) to the start of flowering (BBCH-57) in the highly susceptible cultivar "Carignan"© or before rain events for the other cultivars. The second application period (2–3 sprays on all susceptible cultivars) was from bloom (BBCH-60) to berries, at the "beginning to touch" (BBCH-77) phenological stage. For the third application period, sprays were applied according to disease monitoring on the berries [22]. The details of each trial are given in Table 2.

The fungicides in the experimental trials were sprayed with a Turbo 400 gun sprayer (100-L, 1400 kPa, Degania Sprayers, Degania, Israel) at spray volumes of 800–1000 L/ ha, according to the size of the vines, at time intervals specified for each trial on a phenological and primary infection basis. Unsprayed vines served as the control.

The treatments within the experiments for all the field trials in Chile and Israel were arranged in a randomized complete block design. Plots consisting of 6–10 adjacent vines were replicated 4 times. Disease evaluations were conducted on 4–5 central vines to ensure the prevention of drift from adjacent treatments. Unsprayed buffer rows ensured the prevention of drift from adjacent commercial plots.

A total of 2 large-scale demonstration trials were conducted in 2021 using the wine grape cultivar "Carignan"© in the Anava vineyard and the table grape cultivar "Muscat of Alexandria"© at the Latroon vineyard (Table 2). The fungicides were applied at 1000 L/ha with a tractor-mounted, 2000 L trailed turbo-blower (speeder type) sprayer (Degania Sprayers, Israel) on 0.3 ha of each treated plot for each trial. For the controls, 5 vines were left unsprayed at the end of each of the 4 rows. The methods of fertilization, irrigation, and other cultural practices in all the vineyards were as recommended to the growers in the region, and they were equal for all treatments. Basal leaf removal (all leaves up to the first cluster) was conducted on both sides of the vines four weeks after bloom.

### 2.3. Assessment of Powdery Mildew on the Fruit Clusters

Clusters that were naturally infected with powdery mildew fungus were rated at various intervals after the last application of fungicides. A total of 25 clusters were randomly selected from each side of each row of the four center vines in each replicate plot (50 clusters per replicate, 200 clusters per treatment) and rated for disease severity according to the percentage of the infected cluster area (severity). In this way, the percentage of infected clusters with powdery mildew (incidence) was determined.

### 2.4. Data Analysis

All data were analyzed with the JMP statistics package version 14.1 (SAS, Cary, NC, USA). One-way analysis of variance (ANOVA) was applied to the percent of infected clusters (incidence) and to the arcsin-transformed data of the percent of infected cluster area parameters (severity) to achieve a normal distribution. The Tukey–Kramer HSD test was applied to determine whether the differences between the treatments were significant at $p < 0.05$.

## 3. Results

### 3.1. Field Trials in Chile

Severe powdery mildew disease was observed in clusters of the susceptible cultivars in Chile during the 2017–2018 season. The disease incidence on clusters of both cultivars ranged from 93 to 100%, and the disease severity ranged from 43 to 47% on the untreated control vines. Difenoconazole-TTO effectively inhibited powdery mildew development on clusters in both trials, and it provided 95–99% efficacy in reducing disease incidence and 81.7 and 99.8% efficacy in disease severity compared with the untreated controls (Table 3). In the first trial on "Crimson Seedless"©, the application of difenoconazole-TTO at both rates was significantly more effective than the standard premixed fungicide (Bellis-containing boscalid plus pyraclostrobin) at reducing disease incidence, and there were no

differences in the disease severity for both the "Chardonnay"© and "Crimson Seedless"© cultivars (Table 3).

**Table 3.** Efficacy of tea tree oil, fungicidal mixtures, and difenoconazole-TTO for the management of grape powdery mildew in the field studies during the 2017–2018 season in Chile.

| Treatment and Rate (g/ha) a.i [a] | Requinoa, Cacachopal Valley cv. "Crimson Seedless"© | | Casablanca cv. "Chardonnay"© | |
|---|---|---|---|---|
| | Incidence (%) | Severity (%) | Incidence (%) | Severity (%) |
| Control | $100.0 \pm 0.0$ a [b] | $47.0 \pm 0.2$ a | $93.0 \pm 0.6$ a | $43.0 \pm 0.3$ a |
| Difenoconazole-TTO 60 and 120 | $9.0 \pm 0.3$ c | $0.1 \pm 0.0$ b | $22.0 \pm 0.4$ b | $0.1 \pm 0.1$ b |
| Difenoconazole-TTO 80 and 160 | $5.0 \pm 0.2$ c | $0.1 \pm 0.0$ b | $17.0 \pm 0.4$ b | $0.3 \pm 0.1$ b |
| Boscalid-pyraclostrobin 252 and 128 | $23.0 \pm 0.4$ b | $1.0 \pm 0.0$ b | $27.0 \pm 0.5$ b | $0.3 \pm 0.1$ b |

[a], in the Requinoa vineyard, 6 foliar sprays were applied at an average spray volume of 1200 L/ha at 10-day intervals from the end of bloom (BBCH 69) on 27 November 2017, and then again on 7, 17, and 27 December 2017 and 6 January 2018, and once more when all berries were touching before veraison (BBCH 80) on 16 January 2018. In the Casablanca vineyard, 5 foliar sprays were applied at an average spray volume of 600 L/ha at 14-day intervals from the end of bloom (BBCH 69) on 30 November 2017, and then again on 14 and 28 December 2017 and 11 January 2018, and once more when all berries were touching before veraison (BBCH 80) on 25 January 2018. Disease incidence in both trials was rated seven days after the last application. [b], the numbers represent the means ± SEs of each treatment. The values within the columns that are followed by the same letter are not significantly ($p > 0.05$) different according to the Tukey–Kramer HSD test.

### 3.2. Field Trials in Israel

Severe powdery mildew disease was observed on fruit clusters of the highly susceptible cultivar "Carignan"© in all three trials conducted in the 2020 season in Israel. The disease incidence on clusters of this cultivar ranged from 92 to 99%, and the disease severity ranged from 44 to 98% on the untreated control vines. The trial in Gimzo showed that difenoconazole-TTO provided disease control efficacy measurements of 92.5% and 99% for disease incidence and severity, respectively, compared with the untreated vines, and it was significantly more effective than the standard treatment with tetraconazole (Table 4). In both trials at Eshtaol and the Neta Farm, both difenoconazole-TTO and tetraconazole similarly reduced disease incidence and severity on clusters compared with the untreated vines (Table 4). The trial at the Noham vineyard on the cultivar "Carignan"© in 2021 showed that difenoconazole-TTO, at both rates, provided superior disease control, and its efficacy measurements were 95 and 99% for disease incidence and severity, respectively, when the higher rate was used compared with the untreated vines (Figure 1). Difenoconazole alone was less effective at reducing disease incidence, and it provided only 53.3% efficacy compared with the untreated controls (Table 5). For the table grape cultivar "Red Loosh"© at the Gimzo vineyard, difenoconazole-TTO, at both rates, was significantly more effective than tebuconazole at controlling powdery mildew, providing 98–99% efficacy compared with 76% efficacy in disease incidence by tebuconazole, with no significant difference in disease severity (Table 5). The results of both large-scale demonstration trials conducted in 2021 at the Anava vineyard on the wine grape cultivar "Carignan"© and at the Latroon vineyard on the table grape cultivar "Muscat of Alexandria"© confirmed the high efficacy of difenoconazole-TTO. It provided similar highly effective disease control on fruit clusters as the commercial treatment that included mixtures of leading fungicides, and it provided efficacy measurements of 90 to 95% and 87 to 97% in disease incidence and severity, respectively, for both treatments compared with the untreated control vines (Table 6).

**Table 4.** Efficacy measurements of tetraconazole and difenoconazole-TTO for the management of grape powdery mildew in the field studies on clusters of the cultivar "Carignan"© in 2020 in Israel.

| Treatment and Rate (g/ha) a.i [a] | Gimzo | | Eshtaol | | Neta Farm | |
|---|---|---|---|---|---|---|
| | Incidence (%) | Severity (%) | Incidence (%) | Severity (%) | Incidence (%) | Severity (%) |
| Control | 91.9 ± 3.2 a [b] | 76.7 ± 6.8 a | 98.1 ±9.2 a | 98.3 ± 7.6 a | 98.8 ± 8.6 a | 44.3 ± 6.2 a |
| Difenoconazole-TTO 60 and 120 | 6.9 ± 3.1 c | 0.1 ± 0.02 c | 38.1 ± 3.6 b | 11.4 ± 4.8 b | 10.0 ± 2.5 b | 0.4 ± 0.2 b |
| Tetraconazole 40 | 16.9 ± 2.1 b | 2.5 ± 0.9 b | 31.9± 2.4 b | 5.8 ± 3.4 b | 13.8 ±3.2 b | 0.6 ± 0.4 b |

[a], at the Gimzo vineyard, 5 foliar sprays were completed, starting on 6 April, and again on 21 April, 3 May, and 24 and 6 June 2020. Disease was rated 27 days after the last application at the beginning of veraison. At the Eshtaol vineyard, 5 foliar sprays were completed, starting on 6 April, and again on 23 April, 3 May, and 25 and 8 June 2020. Disease was rated on 18 June 2020 (10 days after the last application). At the Neta Farm, 5 foliar sprays were completed, starting on 4 April, and again on 14 and 21 April, 7 and 15 May, and 3 June 2020. Disease was rated 22 days after the last application at the beginning of veraison. In all trials, the foliar sprays began at 25 cm shoot length and ended when most of the berries were touching (BBCH 79). [b], the numbers represent the means ± SEs of each treatment. The values within the columns that are followed by the same letter are not significantly ($p < 0.05$) different according to the Tukey–Kramer HSD test.

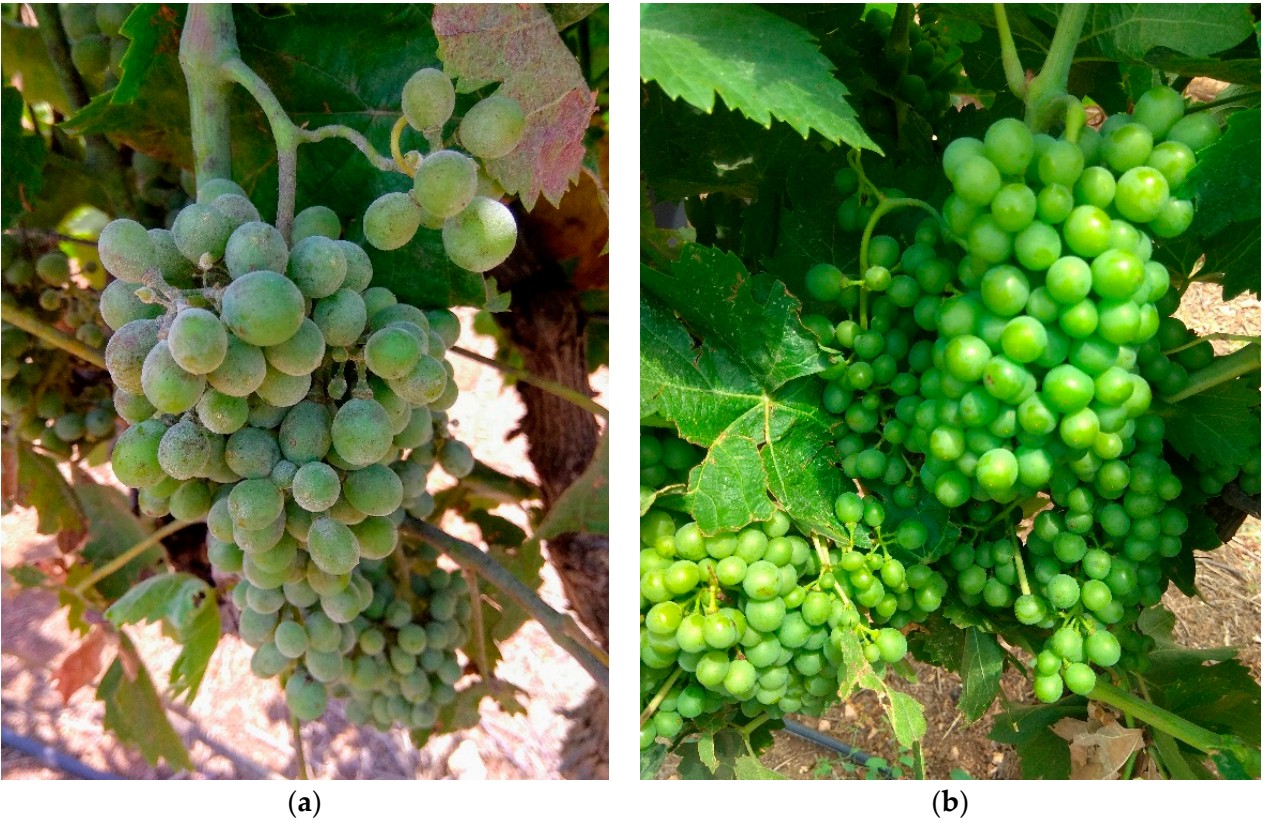

(**a**)  (**b**)

**Figure 1.** Efficacy of difenoconazole-TTO (80 + 160 gr/ha a.i) against grape powdery mildew on cv. "Carignane"© grown in the "Noham" vineyard in 2021. Photos were taken on 8 June 2021 (the last application date). Note the high infection on control untreated (**a**) compared with difenoconazole-TTO treated (**b**).

**Table 5.** Efficacy of difenoconazole, tebuconazole, and difenoconazole-TTO for the management of grape powdery mildew in field studies conducted in 2021 in Israel.

| Treatment and Rate (g/ha) a.i [a] | Noham cv. "Carignane"© | | Gimzo cv. "Red Loosh"© | |
|---|---|---|---|---|
| | Incidence (%) | Severity (%) | Incidence (%) | Severity (%) |
| Control | 100.0 ± 0.0 a [b] | 98.9 ± 3.1 a | 96.3 ± 7.6 a | 16.0 ± 10.1 a |
| Difenoconazole-TTO 40 and 80 | 25.8 ± 6.4 c | 1.6 ± 0.6 b | 1.9 ± 1.1 c | 0.02 ± 0.1 b |
| Difenoconazole-TTO 80 and 160 | 5.0 ± 0.3 d | 0.1 ± 0.1 b | 1.3 ± 0.3 c | 0.02 ± 0.1 b |
| Difenoconazole 37.5 | 46.7 ± 9.1 b | 2.8 ± 2.4 b | n.t | n.t. |
| Tebuconazole 50 | n.t | n.t. | 22.5 ± 9.4 b | 0.3 ± 0.1 b |

[a], at Noham, 5 foliar sprays were completed, starting on 13 April at 15 cm shoot length and again on 29 April, 12 May, and 26 and 8 June 2021. Disease was rated 14 days after the last application at the tight clusters stage. At Gimzo, 4 foliar sprays were completed using table grape vines, starting on 7 May at 80% bloom and again on 20 May, 3 June, and 17 June 2021. Disease was rated 17 days after the last application. In both trials, the sprays ended when a majority of the berries were touching (BBCH 79). [b], the numbers represent the means ± SEs of each treatment. The values within the columns that are followed by the same letter are not significantly ($p < 0.05$) different according to the Tukey–Kramer HSD test.

**Table 6.** Efficacy of difenoconazole-TTO and the tank mixes of fungicides for the management of grape powdery mildew in the large-scale demonstration trials conducted in 2021 in Israel.

| Treatment and Rate (g/ha) a.i [a] | Anava cv. "Carignane"© | | Latroon cv. "Muscat of Alexandria"© | |
|---|---|---|---|---|
| | Incidence (%) | Severity (%) | Incidence (%) | Severity (%) |
| Control | 89.0 ± 8.4 [b] | 15.0 ± 10.4 | 31.5 ± 13.6 | 3.7 ± 6.2 |
| Difenoconazole-TTO 60 and 120 | 7.0 ± 1.2 | 1.5 ± 0.5 | 1.0 ± 0.2 | 0.3 ± 0.2 |
| Commercial standard | 4.4 ± 0.6 | 1.0 ± 0.3 | 1.0 ± 0.1 | 0.5 ± 0.3 |

[a], at Anava, t2 foliar sprays were completed, starting on 5 May at 25% bloom and then again on 19 May 2021. Tank mixtures of cyflufenamid at 5 g.a.i/ha plus quinoxyfen at 50 g.a.i/ha and pyriofenone at 90 g.a.i/ha plus tetraconazole at 40 g.a.i/ha were applied for the first and the second applications, respectively, as standard commercial treatments. Disease was rated 2 June 2021 (14 days after the last application), when most of the berries were touching (BBCH 79). At Latroon, 3 foliar sprays were completed using table grape vines, starting on 12 May at 40% bloom and again on 27 May and 17 June 2021. Tank mixtures of cyflufenamid at 5 g.a.i/ha plus penconazole at 40 g.a.i/ha, pyriofenone at 90 g.a.i/ha plus proquinazid at 40 g.a.i/ha, and meptyldinocarp- at 90.4 g.a.i/ha plus 38.4 g.a.i/ha of myclobutanil were applied for the first, second, and third applications, respectively, as standard commercial treatments. Disease was rated on 4 July 2021 (17 days after the last application), when most of the berries were touching (BBCH 79). In both trials, the sprays were applied with a commercial speeder-type sprayer, with a spray volume of 1000 L/ha on 0.2 ha of each treated plot in each trial. For the controls, five vines were left at the end of each of the four rows. Disease ratings were made on 50 clusters randomly selected from the 4 vine groups in each of the 4 rows of each treatment (i.e., a total of 200 clusters per treatment). [b], the numbers represent the means ± SEs of each treatment.

## 4. Discussion

This study evaluates the efficacy of the first hybrid fungicide containing a combination of a natural product (TTO) with broad-spectrum activity, classified as FRAC Code BM 01 [23], and a traditional site-specific triazole chemical (i.e., difenoconazole) against grape powdery mildew. Our results show that difenoconazole-TTO can effectively control powdery mildew with similar or better efficacy than standard fungicides in two different regions and climates, such as Israel and South America (Chile).

Fungicides with different single-site modes of action (i.e., those belonging to different FRAC codes) are often combined in mixtures to expand the spectrum of activity, prolong persistence, and improve disease control by exploiting additive or synergistic interactions between the components [24–27]. Such mixtures are also frequently recommended for resistance management, which slows the rate of resistance development in the pathogen population, thus extending the effective lives of the fungicides involved.

Tea tree oil, an essential oil, is regarded as a low-risk biopesticide with no known resistance in the target pathogens [23], and thus, no maximum residue levels (MRLs) are required to be established for various crops. TTO is registered by STK Bio-ag Technologies as an active ingredient in USA, Canada, and China, and it is included in Annex I in the EU. Several TTO-based products are registered and in use in more than 30 countries. TTO inhibited spore germination, mycelial growth, lesion development, the formation of reproductive structures, and the sporulation of *Mycosphaerella fijiensis* in banana [28], *Bremia lactucae* in lettuce [29], and *Erysiphe necator* in grapevine [30]. TTO was found to exhibit a strong curative activity against *Mycosphaerella fijiensis* in banana [31], enabling its use even when disease is already visible. A single application of TTO-based product suppressed colonies of powdery mildew on diseased cucumber leaves [31]. Microscopic observations showed that TTO directly affected mycelia and conidia, which collapsed and exhibited irregular shapes. This was accompanied by significant shrinkage of the hyphae, conidia, and conidiophores exposed to TTO [31]. In addition, TTO was found to be an activator of host defense mechanisms, and it systemically induced resistance in banana and tomato plants [32]. TTO enhanced pathogenesis-related (PR) proteins and the expression of marker genes for systemic acquired resistance and induced systemic resistance in both tomato and banana plants [32]. Further studies are needed to demonstrate these activities in grapevines.

In field studies, TTO was found to be effective against a broad range of plant-pathogenic fungi in numerous crops, including vegetables, herbs, banana, and fruit trees [16,33]. Preliminary work has shown that TTO effectively controls grape powdery and downy mildews in laboratory and field studies [30].

As a suitable mixture partner for TTO, a chemical was needed that is stable over time, does not reduce the activity of TTO, and has a similar spectrum of activity against target pathogens. Additionally, like formulated TTO, it should not move systemically in a plant. If TTO were combined with a systemic fungicide such as the DMI triazole flutriafol, the two components would separate and be present in different locations inside a plant's tissues, negating the benefit of a mixture treatment for resistance management. Thus, the non-systemic or limited systemic difenoconazole was chosen as the mixture partner compound for TTO.

The DMI compound difenoconazole inhibits the biosynthesis of ergosterol, a vital fungal cell membrane component. Specifically, DMIs act by inhibiting the cytochrome P-450 sterol 14$\alpha$-demethylase (P-45014DM), which is encoded by CYP51, a key enzyme in the biosynthesis pathway of ergosterol, which is a precursor of a cell membrane component in fungi [23,34–37].

The different modes of action of TTO and difenoconazole render difenoconazole-TTO an effective tool for managing resistance in integrated disease management programs. Furthermore, the combination of a synthetic fungicide with a natural essential oil product results in a reduced synthetic chemical load on the environment, which is highly favored by regulatory agencies and consumers. Difenoconazole-TTO can be alternated with products that exhibit different modes of action and those for which a loss of sensitivity in fungal pathogen populations has been shown. Interspersing fungicide applications with biocontrol agents and low-risk chemicals has been reported to improve powdery mildew control and the shelf-life of grapes [12], and it may help with resistance management. The results of this study show that difenoconazole-TTO was highly effective against powdery mildew in the most susceptible cultivar ("Carignan"©) grown in Israel, where it provided levels of high disease control. Difenoconazole-TTO provided superior disease control and high efficacy measurements of 95% and 99% in disease incidence and severity, respectively, compared with the untreated vines. It was significantly more effective than difenoconazole alone at reducing disease incidence, which provided only 53.3% efficacy compared with the untreated controls (Table 5). Difenoconazole-TTO was more effective than the premixed fungicide (Bellis-containing boscalid plus pyraclostrobin) in Chile (Table 3) or tetraconazole in Israel (Table 4). In commercial applications for the large-scale demonstration trials in

Israel in 2021, difenoconazole- TTO was as effective as the mixtures of fungicides applied in rotation, and it offered >90% efficacy for disease incidence and severity compared with the untreated controls (Table 6).

Difenoconazole-TTO is currently registered in various countries, including the United States, and is effectively used for controlling a broad range of diseases in arable crops including cereals, fruits, nuts, and vegetables [38,39]. With preventive and curative activities, and multiple modes of action including the indirect effect on the host plant by inducing systemic resistance and reduced chemical residues in crops and the environment, difenoconazole-TTO has the potential to be an important component in resistance management and an effective component in the control strategy for grape powdery mildew and various other plant diseases.

## 5. Conclusions

This study focused on efficacy of a new premixed hybrid fungicide containing the demethylation inhibitor (DMI) difenoconazole and essential tea tree oil (TTO), derived from the *Melaleuca alternifolia* plant, against grape powdery mildew in field trials in Chile and Israel. Preventive foliar applications provided up to 99% efficacy in disease incidence and severity compared with the untreated control and were as or more effective than other DMI fungicides, or treatments that include various fungicides applied in rotation or mixtures of fungicides. This new hybrid fungicide can be included in powdery mildew control programs for grapevine, and various other diseases, as a strategic approach in fungicide resistance management.

**Author Contributions:** M.R. and C.J.A. planned the experiments; S.O. performed the trials in Israel; M.R., C.J.A. and S.O. were responsible for data collection and analysis; and M.R. and C.J.A. prepared the manuscript. All authors have read and agreed to the published version of the manuscript.

**Funding:** This research received no external funding.

**Institutional Review Board Statement:** This study does not contain any experiments that were performed by any of the authors with human participants or animal subjects.

**Data Availability Statement:** The data presented in this study are available on request from the corresponding author.

**Acknowledgments:** The authors thank the vineyards personnel in Chile for their valuable assistance and the team of Shmuel Ovadia for their assistance in conducting the field trials in Israel.

**Conflicts of Interest:** The author declares no conflict of interest.

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
