# Peer review of "An Effective Hybrid Fungicide Containing Tea Tree Oil and Difenoconazole for Grape Powdery Mildew Management"

_agriculture, doi:10.3390/agriculture13050979_

Round 1
Reviewer 1 Report
Dear authors,
The manuscript entitled "An Effective Hybrid Fungicide Containing Tea Tree Oil and Difenoconazole for Grape Powdery Mildew Management" presents timely and adequate information to the readers. The introduction is well wriiten and supported by relevant references. The material and methods are well-described and the results are also clearly presented. The discussion presents also adequate information and is suppported by the results.
I have some minor corrections below:
- Line 251: Please clarify this sentence. "in an area with a Mediterranean climate, such as Israel and South America (Chile)." Does Chile harbour a Mediterranean climate?
- Line 325: Remove the double comma in the reference numbers
- Why do you carried out your trials in Chile and Israel. Any particular reason?
- The authors could also add a conclusions section to the manuscript, so can give a better support to the results you obtained
Congrats on your manuscript
Kind Regards
Author Response
Please see the attachment: Response to Reviewer 1

Reviewer 2 Report
The manuscript must be developed. Please see my suggestions.
L30, L32. Repetitive “one of the most important..” and not scientific value statement. Please reshape.
L 40. [5,3,6,7] must be corrected as [5, 5-7]. L42. [8,9,10,6,3,11] references must be in order [3,6,8-11]; L57. Consecutive references [14,15,16,17,18] must be written as (14-18]. Please check the Instructions for authors. Moreover, the references must be inserted near the information provided (antiseptic [ ], bactericide [ ], fungicide [ ]), not as a group. Revise the entire manuscript related to the way of inserting the references.
Discussion. How these fertilizers affect the grape’s composition? A paragraph must be added. I suggest checking and referring to Oprea, et al. Researches on the chemical composition and the rheological properties of wheat and grape epicarp flour mixes, Rev. Chim., 69(1), 2018, 70-75 and https://doi.org/10.3390/life13010178
Nano farming (nano fungicide?) must be reminded as it is the newest direction in agriculture – develop a paragraph – you will find helpful https://doi.org/10.1016/j.chemosphere.2021.132533
Also, a paragraph about the toxicity of the compounds discussed.
Conclusions section is missing. Please add it, underlining the novelty your results brings to the field.
Reviewer 3 Report
Dear Corresponding Author
I checked your paper and I have some suggestions to improve your paper:
1) Regarding my experiences data in such experiments do not follow normal distribution and you will need to transform of data. But after transformation, generally data will not in normal distribution, too. Therefore, you need to analyze your data with non-parametric data.
2) LSD test is just suitable to compare each treatment with control. Therefore, you will need to use multi-comparison test to compare treatments with each other.
3) You have several different locations to evaluate your fungicide in different time frmaes. You will need to use Latin Square in location and time experimental dsign to analyze your data.
4) In Incidence and severity table you have to add standard error as +/-.
5)At least one photo from each location before and after usage of fungicide is needed to add into the paper.
Regards
Round 2
Reviewer 3 Report
Dear Corresponding Author
I checked your paper and I think there is a problem in Table 3. In severity part, standard error is equal and/or higher than average. It shows that you have a problem in your data. Generally, if SE multiplied by 2, result should be lesser than average.
Please check it again and I am waiting to your revised version.
Regards
Author Response
Dera Reviewer,
Thank you for your comment. I rechecked the data for this trial from Chile.
Sorry for my typo mistake. The SE should be 0.1 and not 0.4. I have corrected it in this table.
Thanks again,
Moshe Reuveni
Round 3
Reviewer 3 Report
Dear Corresponding Author
I think the paper is ready to publication now.
Regards
Author Response
Thank for the review